# Synthesis of Halogenated 1,5-Diarylimidazoles and Their Inhibitory Effects on LPS-Induced PGE_2_ Production in RAW 264.7 Cells

**DOI:** 10.3390/molecules26206093

**Published:** 2021-10-09

**Authors:** Zunhua Yang, Yuanying Fang, Jae-Min Kim, Kyung-Tae Lee, Haeil Park

**Affiliations:** 1College of Pharmacy, Jiangxi University of Traditional Chinese Medicine, Nanchang 330004, China; joshyyy@126.com (Z.Y.); fangyuanying@163.com (Y.F.); 2College of Pharmacy, Kyung Hee University, Seoul 02453, Korea; 2014102197@khu.ac.kr (J.-M.K.); ktlee@khu.ac.kr (K.-T.L.); 3College of Pharmacy, Kangwon National University, Chuncheon 24341, Korea

**Keywords:** 1,5-diarylimidazole, halogenation, COX-2, PGE_2_ production, anti-inflammatory

## Abstract

A series of halogenated 1,5-diarylimidazole compounds were synthesized and their inhibitory effects on LPS-induced PGE_2_ production in RAW 264.7 cells were evaluated. A wide variety of 2,4-, 4-, and 2-halogenated 5-aryl-1-(4-methylsulfonylphenyl)imidazoles were synthesized for SAR study via two different pathways. Overall, 4-halogenated 5-aryl-1-(4-methylsulfonylphenyl)imidazoles, regardless of the species of halogen, exhibited very strong inhibitory activities of PGE_2_ production. Among them, 4-chloro-5-(4-methoxyphenyl)-1-(4-methylsulfonylphenyl)imidazole (**3**, IC_50_ 3.3 nM ± 2.93), and 4-chloro-5-(4-chlorophenyl)-1-(4-methylsulfonylphenyl)imidazole (**13**, IC_50_ 5.3 nM ± 0.23) showed the best results.

## 1. Introduction

Cyclooxygenase (COX), is one of the key proinflammatory enzymes which catalyzes the conversion of arachidonic acid to prostaglandins (PGs). Cyclooxygenase exists in two isoforms, COX-1 and COX-2. Among them, COX-2 is inducible and known as a major isoform found in inflammatory lesions [1].

During the past two decades, extensive efforts have been made on the development of selective COX-2 inhibitors by modifying the central heterocycle scaffold of tricyclic lead compounds. A wide variety of 5-member heterocycle scaffolds can serve as scaffolds for COX-2 inhibitors, such as pyrazole (Celecoxib), thiazole (DUP 697), furanone (Rofecoxib), isoxazole (Valdecoxib), imidazole (Cimicoxib), and pyrrole (Figure 1) [2,3,4,5,6,7,8,9,10]. It was well acknowledged from previous SAR studies that the nature of the central heterocycle scaffold is very important for the bioactivity as well as selectivity. Therefore, the design of new compounds based on alternative structural scaffolds has been demanded.

We previously reported a series of 1,5-diarylimidazole analogs, along with their inhibitory activity toward COX-2 enzyme [11,12,13]. Through a series of studies, we found that 1,5-diarylimidazole analogs with the 4-methylsulfonylphenyl group at 1- or 5-position are important for their inhibitory activities against COX-2 catalyzed PGE_2_ production, but its position can be exchanged without significant reduction of bioactivity. We also observed that halogen substitution at the 2- or 4-position of 1,5-diarylimidazole ring significantly influenced bioactivity. These results led us to conduct a supplementary SAR study of 1,5-diarylimidazoles halogenated at 2- or/and 4-position(s) of the imidazole ring (Figure 2).

## 2. Results and Discussion

### 2.1. Synthesis of Halogenated 5-Aryl-1-(4-methylsulfonylphenyl)imidazoles Analogs

A series of 5-aryl-1-(4-methylsulfonylphenyl)imidazoles with halogen(s) on the imidazole ring were synthesized to conduct further SAR study. 5-Aryl-1-(4-methylsulfonylphenyl)imidazoles (**1** R=OCH_3;_
**11** R=Cl) were selected since these compounds exhibited strong inhibitory activities of PGE_2_ production in the precedent experiment [11,12,13].

The target halogenated 1,5-diarylimidazole analogs were prepared following the procedures and conditions as shown in Figure 1. 5-Aryl-1-(4-methylthiophenylphenyl)imidazoles (**Ia** and **Ib**), the key starting materials, were synthesized by 1,3-dipolar cycloaddition of 4-methylsulfanylbenzylidenearylamines and tosylmethyl isocyanide (TosMIC) in the presence of K_2_CO_3_. Following oxidations of **Ia** and **Ib** with oxone^TM^ afforded 5-aryl-1-(4-methylsulfonylphenyl)imidazoles (**1** and **11**, respectively) as described in our previous article [12,13]. Reactions prepared by reacting the 5-aryl-1-(4-methylthiophenylphenyl)imidazoles (**Ia** and **Ib**) with halogenating reagents (NCS, NBS) in CHCl_3_ or CH_3_CN (NIS) reflux conditions (Halogenation ′′A′′) for 2–5 h yielded products as mixtures, and then purification by silica gel column chromatography afforded pure 2,4- and 4-halogenated 5-aryl-1-(4-methylsulfonylphenyl)imidazole analogs [14]. Halogenations of 5-aryl-1-(4-methylsulfonylphenyl)imidazoles at 2-position were not successful with NCS and NBS. 2-Chloro- and 2-bromo-5-aryl-1-(4-methylsulfonylphenyl)imidazole analogs (**4**, **7**, **14**) were prepared by the alternative procedure (Halogenation “B”) from 5-aryl-1-(4-methylthiophenyl)imidazoles (**Ia** and **Ib**) in two steps. Reaction of **Ia** and **Ib**, halogenating reagents (NCS, NBS), and lithium bis(trimethylsilyl)amide (LiHMDS) in THF followed by oxidation with 3-chloroperbenzoic acid yielded products. Thus, eighteen 5-aryl-1-(4-methylsulfonylphenyl)imidazoles with halogen(s) (Cl, Br, I) at 2/4-, 4-, 2-position(s) were synthesized via two different synthetic procedures and results are shown in Table 1.

### 2.2. RAW 264.7 Cell Culture and Measurement of PGE_2_ Concentrations

RAW 264.7 cells obtained from the American Type Culture Collection (ATCC, VA, USA) were cultured in a petri dish in DMEM supplemented with 10% FBS and 1% antibiotics under 5% CO_2_ at 37 °C for 3 days based on the previously described procedures [15]. Briefly, cells were plated in 96-well plates (2 × 10^5^ cells/well). After pre-incubation with the test compounds for 1 h, LPS (1 μg/mL) were added and incubated for 24 h. PGE_2_ concentration in the medium was measured using an ELISA kit for PGE_2_ (Cayman Chem. Co.) according to the manufacturer’s recommendation.

Cell viability was assessed with MTT assay. All tested compounds showed no or less than 10% reduction of MTT assay at the tested concentrations, indicating that they were not significantly cytotoxic to RAW 264.7 cells in the presence or absence of LPS (Appendix A). Therefore, the inhibition of PGE_2_ production by halogenated 5-aryl-1-(methylsulfonylphenyl)imidazoles might not be associated with their cytotoxicity.

The inhibitory activities of 5-aryl-1-(4-methylsulfonylphenyl)imidazoles (**1** and **11**) and their halogenated analogs (**2**~**10**, **12**~**20**) on PGE_2_ production against LPS-induced RAW 264.7 cells were estimated and results are shown in Figure 3, Figure 4 and Figure 5 and Table 1.

As demonstrated in Figure 3 and Figure 4 and Table 1, several halogenated 5-aryl-1-(4-methylsulfonylphenyl)imidazoles (**3**, **7**, **9**, **13**, **16**, **19**) exhibited remarkable inhibitory activities of PGE_2_ production. All the 2,4-dihalogenated 5-aryl-1-(4-methylsulfonylphenyl)imidazole analogs (**2**, **5**, **8**, **12**, **15**, **18**) showed reduced inhibitory activities of PGE_2_ production. Among them, 2,4-dihalogenated (Cl, Br) analogs (**12**, **15**) of the parent compound **11** showed much reduced inhibitory activities than analogs (**2**, **5**) from the parent compound **1**. 2,4-Diiodinated analogs (**8**, **18**) showed slightly more reduced inhibitory activities of PGE_2_ production than their parent compounds **1** and **11**, respectively.

Synthesized 4-halogenated 5-aryl-1-(4-methylsulfonylphenyl)imidazole analogs mostly exhibited very strong inhibitory activities of PGE_2_ production regardless of the parent compound and the species of halogen as we observed from those of compounds **3**, **9**, **13**, **16** and **19** in Figure 5 and Table 1. Among them, 4-chloro-5-(4-methoxyphenyl)-1-(4-methylsulfonylphenyl)imidazole (**3**, IC_50_ 3.3 nM ± 2.93) and 4-chloro-5-(4-chlorophenyl)-1-(4-methylsulfonyl-phenyl)imidazole (**13**, IC_50_ 5.3 nM ± 0.23) showed best inhibitory activities of PGE_2_ production.

The inhibitory activity of PGE_2_ production of 2-halogenated 5-aryl-1-(4-methylsulfonylphenyl)imidazoles is somewhat complicated to predict a tendency. In case of 5-(4-methoxyphenyl)-1-(4-methylsulfonyl-phenyl)imidazole (**1**) as the parent compound, mono-halogenation (Br, I) at 2-position exhibited much better inhibitory activity of PGE_2_ production (**7** and **10**, respectively), whereas 2-chlornation showed much reduced inhibitory activity of PGE_2_ production (**4**) compared to that of the parent compounds (**1**). In case of 5-(4-chlorophenyl)-1-(4-methylsulfonylphenyl)imidazole (**11**) as the parent compound, mono-halogenation (Cl, Br) at 2-position resulted in completely loss of inhibitory activity of PGE_2_ production (**14** and **17**, respectively), while 2-iodination showed equal inhibitory activity of PGE_2_ production (**20**) compared to that of the parent compounds (**11**).

To examine the selectivity of halogenated imidazole analogs towards COX-2 in comparison to COX-1, we conducted COX Inhibitor Screening Assay (Cayman, MI, USA) using compound **3**, one of lead compounds. We evaluated its potency and selectivity of inhibition in vitro. It was found that compound **3** showed a significant inhibitory effect on COX-2 activity at 100 nM, whereas it did not show any inhibitory effect on COX-1 (Figure 6).

## 3. Conclusions

In our study, a series of halogenated 1,5-diarylimidazole compounds were designed, synthesized, and their inhibitory effects on LPS-induced PGE_2_ production in RAW 264.7 cells were evaluated. Halogenation of 5-aryl-1-(4-methylsulfonylphenyl)imidazoles with NIS, NBS, and NIS afforded diverse 2,4-, 4-, and 2-halogenated 5-aryl-1-(4-methylsulfonylphenyl)imidazoles for SAR study. New synthetic compounds were evaluated for their inhibitory activities on LPS-induced PGE_2_ production in RAW 264.7 macrophage cells. Among them, 4 compounds (**3**, **7**, **13**, **16**) were identified as more potent PGE_2_ production inhibitors than celecoxib (PGE_2_ IC_50_ = 8.7 nM ± 1.35). Most 4-halogenated 5-aryl-1-(4-methylsulfonyl-phenyl)imidazoles, regardless of the species of halogen, exhibited very strong inhibitory activities of PGE_2_ production from LPS-induced RAW 264.7 cells. Overall, the inhibitory activity of PGE_2_ production was largely dependent on the substitution position (4- >> 2- and 2,4-) and size (Cl >> Br > I) of the halogen atom(s). Compounds **3** (IC_50_ 3.3 nM ± 2.93) and **13** (IC_50_ 5.3 nM ± 0.23) were selected as lead candidates for the further study. We are now investigating the molecular target including COX-2 and mPGES-1 and its molecular mechanism to inhibit the LPS-induced PGE_2_ production in macrophages, and the results will be reported in due time.

## 4. Materials and Methods

### 4.1. General

All chemicals, solvents, and reagents were obtained from commercial suppliers and used without further purification, unless specified. All solvents used for reaction were freshly distilled from proper dehydrating agent under nitrogen gas. Reactions were monitored by thin-layer chromatography performed on glass-packed silica gel plates (60F-254) (Merck, Darmstadt, Germany) with UV light. Flash column chromatography was performed with silica gel (100–200 mesh). ^1^H-NMR (300 MHz) was recorded on Bruker DPX 300 spectrometers, (Bruker, Billerica, MA, USA) and ^13^C-NMR was recorded on Bruker Avance Neo 600 spectrometers (Bruker, Billerica, MA, USA), fully decoupled, and chemical shifts are reported in parts per million (ppm) downfield relative to tetramethylsilane as an internal standard. Peak splitting patterns are abbreviated as s (singlet), br s (broad singlet), d (doublet), t (triplet), q (quartet), dd (doublet of doublet), and m (multiplet). Low-resolution mass spectra (LRMS) were recorded on an API 3200 MS system of AB SCIEX. (AB Sciex, Flamingham, MA, USA) High resolution mass spectra (HRMS) were recorded on JEOL JES-X320 electron spin resonance spectrometer. Analytical thin-layer chromatography (TLC) was performed using a commercial glass plate with silica gel 60F254 purchased from Merck. Chromatographic purification was carried out by flash chromatography using Kieselgel 60 (230–400 mesh, Merck, Darmstadt, Germany).

#### 4.1.1. General Procedure for Synthesis of Intermediates **Ia** and **Ib**

To the solution of imine (4 mmol) in methanol (MeOH) and dimethoxyethane (20 mL, *v/v* = 1/2) was added anhydrous potassium carbonate (1.66 g, 12 mmol) and tosylmethyl isocyanide (0.94 g, 4.8 mmol). The reaction mixture was refluxed for 1 day. The solvent was removed, and the residue was extracted with dichloromethane (DCM). The organic layer was washed with brine, dried over magnesium sulfate and concentrated. The residue was purified by silica gel column chromatography with hexane-ethyl acetate (EtOAc) yielded the corresponding imidazole intermediates, respectively.
5-(4-methoxyphenyl)-1-(4-methylthiophenyl)imidazole (**Ia**). Yellow solid, 10%; ^1^H-NMR (300 MHz, CDCl_3_) δ 7.65 (s, 1H, 2-H), 7.24 (d, *J* = 8.6 Hz, 2H, Ar-H), 7.18 (s, 1H, 4-H), 7.08 (m, 4H, Ar-H), 6.81 (d, *J* = 8.8 Hz, 2H, Ar-H), 3.79 (s, 3H, OCH_3_), 2.50 (s, 3H, SCH_3_).5-(4-chorophenyl)-1-(4-methylthiophenyl)imidazole (**Ib**) Yellow solid, 49%; ^1^H-NMR (300 MHz, CDCl_3_) δ 7.67 (d, *J* = 1.0 Hz, 1H, 2-H), 7.30–7.20 (m, 5H, Ar-H, 4-H), 7.07 (m, 4H, Ar-H), 2.51 (s, 3H, CH_3_).

#### 4.1.2. General Procedure for Synthesis of Parent Compounds **1** and **11**

To the solution of intermediates (**Ia** and **Ib**, 1 mmol) in DCM (10 mL) was added, at 0 °C, 3-chloroperbenzoic acid (0.56 g, 2.5 mmol). The mixture was stirred for 2 h, followed by the addition of more DCM, washing with aqueous Na_2_S_2_O_3_, NaHCO_3_, and brine, drying over magnesium sulfate, and concentration. The residue was purified by silica gel column chromatography with hexane-EtOAc.
1-(4-methylsulfonylphenyl)-5-(4-methoxyphenyl)imidazole (**1**). White solid, 84%; ^1^H-NMR (300 MHz, CDCl_3_) δ 7.95 (d, J = 8.6 Hz, 2H, Ar-H), 7.74 (s, 1H, 2-H), 7.38 (d, *J* = 8.6 Hz, 2H, Ar-H), 7.22 (s, 1H, 4-H), 7.05 (d, *J* = 8.8 Hz, 2H, Ar-H), 6.84 (d, *J* = 8.8 Hz, 2H, Ar-H), 3.80 (s, 3H, OCH_3_), 3.08 (s, 3H, SO_2_CH_3_); ESIMS: *m/z* [M + H]^+^ 329.15-(4-chorophenyl)-1-(4-methylsulfonylphenyl)imidazole (**11**). Yellow solid, 87%; ^1^H-NMR (300 MHz, CDCl_3_) δ 8.02 (d, *J* = 8.6 Hz, 2H, Ar-H), 7.78 (s, 1H, 2-H), 7.30 (m, 5H, Ar-H, 4-H), 7.05 (d, *J* = 8.5 Hz, 2H, Ar-H), 3.11 (s, 3H, CH_3_).

#### 4.1.3. General Procedure for Synthesis of Halogenated 5-Aryl-1-(4-methysulfonylphenyl)imidazole Analogs 3, 13, and 16 (Check Supplementary Materials for Other Compounds)

To the solution of compound **1** or **11** (0.5 mmol) in CHCl_3_ (4 mL) was added NCS/NBS/NIS (0.75 mmol). The mixture was refluxed for 5 h, extracted with DCM, washed with aqueous NaHSO_3_ and brine, dried over magnesium sulfate, and concentrated. The residue was purified by silica gel column chromatography to give 4-halo and 2,4-dihalo imidazoles. Different from chlorination, 2-bromo imidazole products were separated in some bromination reactions. While iodination with NIS in acetonitrile (CH_3_CN) afforded 2-, 4- and 2,4-iodo imidazole products. NCS/NBS/NIS are abbreviated name of *n*-chlorosuccinimide, *n*-chlorosuccinimide, and *n*-chlorosuccinimide, respectively.
4-chloro-1-(4-methylsulfonylphenyl)-5-(4-methoxyphenyl)imidazole (**3**). White solid, 41%; ^1^H-NMR (300 MHz, CDCl_3_) δ 7.95 (d, *J* = 8.6 Hz, 2H, Ar-H), 7.65 (s, 1H, 2-H), 7.32 (d, *J* = 8.6 Hz, 2H, Ar-H), 7.11 (d, *J* = 8.8 Hz, 2H, Ar-H), 6.88 (d, *J* = 8.8 Hz, 2H, Ar-H), 3.82 (s, 3H, OCH_3_), 3.09 (s, 3H, SO_2_CH_3_); ^13^C-NMR (150 MHz, CDCl_3_) δ 159.9, 140.6, 135.0, 129.1, 129.0, 127.1, 125.7, 118.7, 114.4, 55.3, 44.4; HRMS (EI) *m*/*z* Calcd for C_17_H_15_ClN_2_O_3_S [M] + 362.0492 Found 362.0493.4-chloro-5-(4-chorophenyl)-1-(4-methylsulfonylphenyl)imidazole (**13**). Yellow solid, 20%; ^1^H-NMR (300 MHz, CDCl_3_) δ 8.00 (d, J = 8.6 Hz, 2H, Ar-H), 7.67 (s, 1H, 2-H), 7.32 (m, 4H, Ar-H), 7.13 (d, *J* = 8.5 Hz, 2H, Ar-H), 3.10 (s, 3H, CH_3_); ^13^C-NMR (150 MHz, CDCl_3_) δ140.8, 140.1, 135.9, 135.2, 131.0, 129.3, 129.2, 125.8, 124.9, 55.1, 49.2, 44.4; HRMS (EI) *m*/*z* Calcd for C_16_H_12_Cl_2_N_2_O_2_S [M] + 365.9997 Found 365.9994.4-bromo-5-(4-chorophenyl)-1-(4-methylsulfonylphenyl)imidazole (**16**). Yellow solid, 50%; ^1^H-NMR (300 MHz, CDCl_3_) δ 7.98 (d, *J* = 8.7 Hz, 2H, Ar-H), 7.72 (s, 1H, 2-H), 7.33 (m, 4H, Ar-H), 7.16 (d, *J* = 8.6 Hz, 2H, Ar-H), 3.10 (s, 3H, CH_3_); ^13^C-NMR (150 MHz, CDCl_3_) δ 140.5, 140.3, 137.1, 135.0, 131.2, 129.2, 129.2, 128.6, 125.8, 125.7, 117.6, 44.4; HRMS (EI) *m*/*z* Calcd for C_16_H_12_BrClN_2_O_2_S [M] + 409.9491 Found 409.9490.

#### 4.1.4. Alternative Procedure for Synthesis of 2-Halogenated 5-aryl-1-(4-methylsulfonylphenyl)imidazole Analogs (Check Supplementary Materials for Other Compounds)

To the solution of 5-aryl-1-(4-methylthiophenyl)imidazoles (**Ia** or **I**, 0.4 mmol) in tetrahydrofuran (THF) 3 mL of lithium bis(trimethylsilyl)amide (1 M in THF, 1.2 mL) was added dropwise at −20 °C. The mixture was stirred for 0.5 h, then solution of NCS or NBS (1.6 mmol) in THF (3 mL) was added. The reaction mixture was stirred for 0.5 h at −20 °C and 6 h at room temperature. Saturated aqueous NH_4_Cl was added to the mixture and extracted with ethyl acetate. The organic layer was washed with aqueous NaHSO_3_ and brine, dried over magnesium sulfate, and concentrated under vacuum. Following oxidation of crude 2-halogenated 5-aryl-1-(4-methylthiophenyl)imidazole with 3-chloroperbenzoic acid followed by silica gel column chromatography yielded pure 2-halogenated 5-(4-methoxyphenyl)-1-(4-methylsulfonylphenyl)imidazoles.
2-bromo-5-(4-methoxyphenyl)-1-(4-methylsulfonylphenyl)imidazole (**7**). White solid, 12%; ^1^H-NMR (300 MHz, CDCl_3_) δ 7.96 (d, *J* = 8.7 Hz, 2H, Ar-H), 7.64 (s, 1H, 4-H), 7.32 (d, *J* = 8.7 Hz, 2H, Ar-H), 7.12 (d, *J* = 8.8 Hz, 2H, Ar-H), 6.88 (d, *J* = 8.8 Hz, 2H, Ar-H), 3.82 (s, 3H, OCH_3_), 3.09 (s, 3H, SO_2_CH_3_); ^13^C-NMR (150 MHz, CDCl_3_) δ 159.9, 140.7, 140.1, 136.4, 131.3, 129.6, 129.0, 125.7, 119.3, 116.9, 114.3, 55.3, 44.4; HRMS (EI) *m*/*z* Calcd for C_17_H_15_BrN_2_O_3_S [M] + 405.9987 Found 405.9988.

### 4.2. Biology

#### 4.2.1. Cell Culture and Sample Treatment

RAW 264.7 cells were purchased from the Korea Cell Line Bank (Seoul, South Korea) and maintained in DMEM medium containing 10% FBS, streptomycin sulfate, penicillin, HEPES, and sodium bicarbonate in a 5% CO_2_ atmosphere at 37 °C. RAW 264.7 cells were incubated with SAMPLE (1, 10, 100 nM and 1, 10 μM) for 1 h and then activated with LPS (100 ng/mL) for the indicated time. SAMPLE were dissolved in DMSO and added to the culture media in serial dilution (the final concentration of DMSO in all experiments did not exceed 0.05%).

#### 4.2.2. MTT Assay for Cytotoxicity

RAW 264.7 cells, 2 × 10^5^ cells/mL with 10% fetal bovine serum cell culture medium, and 1 mL of cell suspension were added to each hole of a 24-well plate for 24 h in a 37 °C incubator. Cytotoxicity was determined by MTT assay after treating various concentrations of SAMPLE (1, 10, 100 nM and 1, 10 μM) for another 24 h after overnight incubation. The formed formazan crystals in the cells were dissolved by DMSO, followed by measurement at 540 nm.

#### 4.2.3. Measurement of NO and PGE_2_ Production

RAW 264.7 cells (2 × 10^5^ cells/mL) were seeded onto a 24-well plate and then incubated with/without LPS (100 ng/mL) in the presence or absence of SAMPLE (1, 10, 100 nM and 1, 10 μM) for 24 h. Nitrite levels of cellular supernatants were measured using the Griess reaction and estimated to reflect the concentration of NO. The absorbance was measured at the wavelength of 540 nm using the microplate reader. In all experiments, fresh culture media were used as blanks. The levels of nitrite in the samples were determined using the standard curve of sodium nitrite. PGE_2_ concentration in the medium was measured using an ELISA kit for PGE_2_ (R&D Systems, Minnesota, MN, USA).

#### 4.2.4. Effects of Samples on the COX-1 and COX-2 Activity

Samples were evaluated for its potency and selectivity of inhibition in vitro using COX Inhibitor Screening Assay (Cayman, Michigan, MI, USA). Recombinant COX-1 (ovine) or COX-2 (human) proteins were pre-incubated with Compound 3 for 10 min at 37 °C. The reaction was started by the addition of 100 µM arachidonic acid and allowed to proceed for 2 min. The reaction was terminated by the addition of an HCl solution containing SnCl_2_. The COX activity assay directly measures PGF2α produced by SnCl_2_ reduction of COX-derived PGH_2_. The prostanoid product is quantified via EIA. As control inhibitors for COX-1 or COX-2, SC-560 (100 nM) or Dup-697 (100 nM) were used.

#### 4.2.5. Statistical Analysis

Results are expressed as the mean ± SD of triplicate experiments with similar patterns. Statistically significant values were compared using ANOVA and Dunnett’s post hoc test, and P values of less than 0.05 were considered statistically significant. # *p* < 0.05 compared with the control group, and * *p* < 0.05, ** *p* < 0.01, and *** *p* < 0.001 compared with the LPS-stimulated group.

## Data Availability

The data presented in this study are available on request from the corresponding author.

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
