# Peer review of "Synthesis of Halogenated 1,5-Diarylimidazoles and Their Inhibitory Effects on LPS-Induced PGE2 Production in RAW 264.7 Cells"

_molecules, 2021, doi:10.3390/molecules26206093_

Round 1

Reviewer 1 Report

The article describes Synthesis of halogenated 1,5-diarylimidazoles and their inhibi- 2 tory effects on LPS-induced PGE2 production in RAW 264.7 3
cells. The paper can be published after several improvements:

Authors should add the NMR data to the Supplementary Materials.

It will be also sufficient to add the IR data of the synthesised compounds.

It will be good to compare the results of the biological activity of synthesised compounds with the results for compounds obtained by other Authors.

Author Response

Corrections-1 (molecules-1406342)

The article describes Synthesis of halogenated 1,5-diarylimidazoles and their inhibitory effects on LPS-induced PGE2 production in RAW 264.7 3 cells. The paper can be published after several improvements:

Q1/ Authors should add the NMR data to the Supplementary Materials.

  • I only have texted NMR data for most compounds. Unfortunately, I lost spectra related with this project during my lab moving in 2020. I have 13C-NMR and HRMS for only more active compounds (3, 7, 13, 16). I collected these data again in the July, 2021.

Q2/ It will be also sufficient to add the IR data of the synthesized compounds.

  • It will be more clear whether halogenation occurred or not if we check IR data. Instead, we collected mass data to identify halogenated products.

Q3/ It will be good to compare the results of the biological activity of synthesized compounds with the results for compounds obtained by other authors.

  • I agreed to the reviewer’s suggestion. We compared the activity of our compounds with that of celecoxibTM, the 1st and sole drug in this market.

Reviewer 2 Report

The manuscript describes synthesis and a brief SAR study of a number of halogenated 1,5-diarylimidazoles as potential COX-2 inhibitors through inhibition of LPS-induced PGE2 production in RAW 264.7 cells.

The study is very limited as it only focuses on COX-2 inhibitory activity and doesn’t explore the selectivity of the synthesized compounds by comparing their inhibition to COX-1. Also, some anti-inflammatory activity would have been essential to link between the in-vitro and in-vivo activity.

The manuscript needs English language revision, some of the mistakes noticed can be seen below:

Line 31: can serve

Line 38: series of studies

Line 41: Exchanges with each other

Line 59: R=OCH3

Line 99: might not be

Line 141: is complicated

Some minor correction is required:

  1. Table 1: What is the calculated standard deviation of your IC50 values?
  2. It would have been interesting if the authors could expand their halogenated derivatives to include the fluorinated ones. They are expected to have better activity.
  3. Have you tried to examine the selectivity of your compounds towards COX2 in comparison to COX1? That would have added value and credibility to your study.
  4. Please expand your list of citation to include original research when referring to active compounds e.g. celecoxib. Also, several other 1,5-diaryl imidazoles have been previously reported as COX-2 inhibitors, the authors need to reference these studies.

Author Response

Corrections-2(molecules-1406342)

*Most errors were corrected:

The manuscript needs English language revision, some of the mistakes noticed can be seen below:

Line 31: can serve

Line 38: series of studies

Line 41: Exchanges with each other

Line 59: R=OCH3

Line 99: might not be

Line 141: is complicated

Some minor correction is required:

  1. Table 1: What is the calculated standard deviation of your IC50 values? à We added the new data including standard deviation in Table 1 of the revised manuscript.

  1. It would have been interesting if the authors could expand their halogenated derivatives to include the fluorinated ones. They are expected to have better activity. à I think so. But fluorination condition is quite different from other halogenation.

  1. Have you tried to examine the selectivity of your compounds towards COX2 in comparison to COX1? That would have added value and credibility to your study. à To follow your comment, we have done the COX-1 and -2 activity assay of compound 3. It was found that compound 3 showed significant inhibitory effects on COX-2 activity at 100 nM, whereas it did not show any inhibitory effect on COX-1. We are now investigating the molecular target including COX-2 and mPGES-1 and its molecular mechanism to inhibit the LPS-induced PGE2 production in macrophages (Figure 6).

Reviewer 3 Report

The article in question synthesizes a series of halo- and dihalo-substituted 1,5-diarylimidazoles. The discussion of this section is confusing, and could be clarified by including a table of the synthesized derivatives with yields and which synthetic method was used. Are the mono- and dihalo-substituted species being isolated from a single reaction? If so, this doesn’t seem to be a very good method for future synthesis of related molecules. This may also explain why a table with yields was not included, as they generally are poor. Halogenation of the sulfonyl-substituted substrate is not surprising as this substituent significantly deactivates the imidazole ring and limits reactivity with the electrophilic halogen species. Oxidation post halogenation is the clear solution to this issue, but it still could be described more efficiently along with a scheme that makes these synthetic challenges more obvious.

The SAR studies are done on a cell line very familiar to the authors from their previous work. It does appear that some of these halogenated derivatives have enhanced bioactivity, and there are some general trends that can be identified including the optimal site for halogenation and size of the atom. I wonder if the size is the only relevant variable, or do electronics play a role in this activity trend as well? In table 1, X1 and X2 are not defined. Presumably these are the 2- and 4- positions on the imidazole ring, but which is which. These could be labeled on Scheme 1 to make it definitive. Also, the data in Figures 3, 4, and 5 are too small. I was reading this draft on paper, and it was very hard to interpret these graphs. Maybe those reading electronically will have an easier time by magnifying the image. Celecoxib is represented in Table 1, but there isn’t much discussion of how these derivatives compare. Most of the discussion focuses on comparing these new derivatives to 1 or 11. In other words, how promising are the lead candidates as drug targets compared to the current industry standard? 

There are a number of typos in the paper, and moderate English grammatical issues that make reading the paper a challenge. The text should be reviewed and rewritten by a native English speaker. Below are some of the typos and errors I noticed:

Line 59: OCH3 Subscript 3 missing

Line 197: imine? Shouldn’t this be the imidazole at this point?

Lines 224, 230, 235: Numbers in the molecular formula should be subscripted

Author Response

Corrections-3 (molecules-1406342)

1/ The discussion of this section is confusing, and could be clarified by including a table of the synthesized derivatives with yields and which synthetic method was used. à Insert Halogenation “A” & “B” to the scheme 1 to clarify.

2/ Are the mono- and dihalo-substituted species being isolated from a single reaction? If so, this doesn’t seem to be a very good method for future synthesis of related molecules. This may also explain why a ------------- issue, but it still could be described more efficiently along with a scheme that makes these synthetic challenges more obvious. à As the reviewer explained, halogenation yields were mostly low. We needed samples for SAR study asap at that time, so halogenation conditions were not optimized. Alternative procedure (Halogenation “B”) was used only for those which couldn’t be synthesized by the general halogenation method.

3/ The SAR studies are done on a cell line very familiar to the authors from their previous work. It does appear that some of these halogenated derivatives have enhanced bioactivity, and there are some general trends that can be identified including the optimal site for halogenation and size of the atom. I wonder if the size is the only relevant variable, or do electronics play a role in this activity trend as well? à In our previous study, the effects of halogens on bioactivity was not very clear since very limited number of halogenated imidazole analogs were used. In the present study, we found both position(s) and size of halogen are important for strong bioactivity.

4/ In table 1, X1 and X2 are not defined. Presumably these are the 2- and 4- positions on the imidazole ring, but which is which. These could be labeled on Scheme 1 to make it definitive.

à Yes, I modified the scheme 1 and Table 1.

5/ Also, the data in Figures 3, 4, and 5 are too small. I was reading this draft on paper, and it was very hard to interpret these graphs. Maybe those reading electronically will have an easier time by magnifying the image. à I will provide original Figures to the editor.

6/ Celecoxib is represented in Table 1, but there isn’t much discussion of how these derivatives compare. Most of the discussion focuses on comparing these new derivatives to 1 or 11. In other words, how promising are the lead candidates as drug targets compared to the current industry standard? à I will compare celecoxib and our lead compounds. As we can roughly observe from Figures, our lead compounds exhibited equivalent bioactivity compared to that of celecoxib.

7/ There are a number of typos in the paper, and moderate English grammatical issues that make reading the paper a challenge. The text should be reviewed and rewritten by a native English speaker. Below are some of the typos and errors I noticed:

Line 59: OCH3 Subscript 3 missing

Line 197: imine? Shouldn’t this be the imidazole at this point?

Lines 224, 230, 235: Numbers in the molecular formula should be subscripted

Errors were corrected as the reviewer suggested